# FIND: Fine-tuning Initial Noise Distribution with Policy Optimization for Diffusion Models

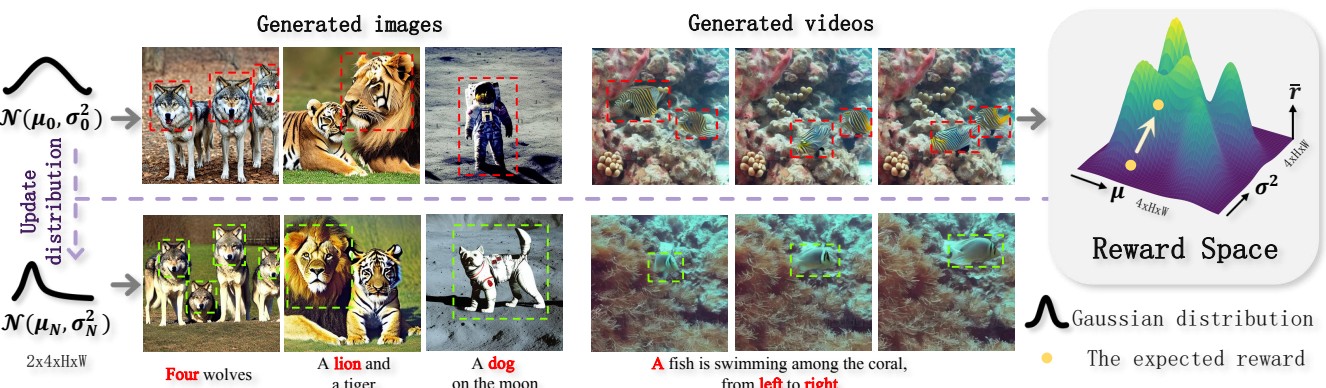

**Figure 1: Our FIND framework optimizes the initial distribution of any diffusion-based model to enhance the consistency between generated content and the prompts provided by users. Before optimization, the semantics of generated images and videos could diverge from the prompt for complex scenes, as indicated by the red boxes. By optimizing the overall expected reward of consistency, the content within the green box becomes consistent with the prompt.**

## ABSTRACT

In recent years, large-scale pre-trained diffusion models have demonstrated their outstanding capabilities in image and video generation tasks. However, existing models tend to produce visual objects commonly found in the training dataset, which diverges from user input prompts. The underlying reason behind the inaccurate generated results lies in the model's difficulty in sampling from specific intervals of the initial noise distribution corresponding to the prompt. Moreover, it is challenging to directly optimize the initial distribution, given that the diffusion process involves multiple denoising steps. In this paper, we introduce a Fine-tuning **I**nitial **N**oise **D**istribution (**FIND**) framework with policy optimization, which unleashes the powerful potential of pre-trained diffusion networks by directly optimizing the initial distribution to align the generated contents with user-input prompts. To this end, we first reformulate the diffusion denoising procedure as a one-step Markov decision process and employ policy optimization to directly optimize the initial distribution. In addition, a dynamic reward calibration module is proposed to ensure training stability during optimization. Furthermore, we introduce a ratio clipping algorithm to utilize historical data for network training and prevent the optimized distribution from deviating too far from the original policy to restrain excessive optimization magnitudes. Extensive experiments demonstrate the effectiveness of our method in both text-to-image and text-to-video tasks, surpassing SOTA methods in achieving consistency between prompts and the generated content. Our method achieves 10 times faster than the SOTA approach.

## CCS CONCEPTS

• **Computing methodologies** → *Computer vision.*

## KEYWORDS

Multimodal Generation, Diffusion Model, Controllable Generation

**ACM Reference Format:**
. 2018. FIND: Fine-tuning Initial Noise Distribution with Policy Optimization for Diffusion Models. In *Proceedings of Make sure to enter the correct conference title from your rights confirmation emai (Conference acronym 'XX).* ACM, New York, NY, USA, 10 pages. https://doi.org/XXXXXXX.XXXXXXX

## 1 INTRODUCTION

Nowadays, the generative capabilities of diffusion models [14, 28, 35] have gained widespread recognition in the domain of text-to-images [27, 28, 30], text-to-videos [11, 38], and text-to-3D [22, 37]. Although diffusion-based generative models excel at creating high-quality content, the semantics of the content generated frequently fail to align closely with the input text prompts. The current model tends to generate highly correlated objects and concepts even with explicit and clear prompts. For example, as shown in Fig. 1, the generated image is still about an astronaut on the Moon with the prompt *'a dog on the moon'*. It remains a significant challenge to ensure consistency between the generated content and the prompt's description.

To address this problem, numerous works [24, 25, 49, 50] are proposed to ensure consistency through the use of additional control

**Unpublished working draft. Not for distribution.**

signals, such as depth maps, edge maps, etc. Although achieving promising results in alignment with users' intentions, these methods require substantial computational resources for large-scale training with auxiliary models. Besides, the provision of additional control mediums is a burden for users. On the other hand, several pioneer methods [3, 10] leverage reinforcement learning to align the generated images with the prompt by fine-tuning network parameters iteratively. The advantage of these approaches is that the output content is approaching to align with the input prompt without needing additional training data. However, due to the sampling nature of reinforcement learning and the extensive optimization of LoRA-like [10] networks, these methods result in longer training times for individual prompts.

Inspired by recent works [4, 8, 43], we observe the essential capability of large-scale pre-trained models to generate diverse and high-quality controllable visual content with a zero-shot fashion. The initial noise distribution significantly impacts the final generated outcomes, influencing aspects such as layout, color, and semantics of the generated content [35]. The reason for the misalignment of the baseline diffusion model partially comes from the sampling bias between the standard normal distribution and the unusual complex prompts provided by users. The training of diffusion utilizes the standard normal distribution, making it easier for the network to generate samples similar to those in the training set when sampling from a normal distribution in testing, as illustrated by the red boxes in the first row of Fig.1. Based on these findings, our motivation is to directly adjust the initial noise to align the generated content with user prompts without any training of the baseline model and extra network structures. After adjusting the initial noise, the baseline model can generate highly aligned images and videos with unconventional prompts, as shown in the second row of Fig.1. However, since diffusion processes require multiple denoising steps, it is challenging to calculate the loss on the final generated result to backpropagate gradients to the initial noise distribution.

In this paper, we propose a novel framework **F**ine-tuning **I**nitial **N**oise **D**istribution (**FIND**) of diffusion model to align the content generated more closely with the input text prompt by adjusting the initial noise distribution with policy optimization. To this end, we formulate the entire optimization process as a one-step Markov decision process and employ policy gradients to optimize the initial distribution. The proposed approach allows the baseline model to bypass the multiple intermediate denoising steps and directly optimize the initial distribution based on the reward derived from the final generated outcome. To ensure the accuracy of the optimization direction for the parameters of the initial distribution, we introduce a dynamic reward calibration module to predict the expected reward of the current initial distribution. As shown on the right side of Fig.1, we need to optimize our initial noise distribution to increase the expected value of the reward without compromising generative performance. The optimization process is then guided by the difference between the reward of sampled data and the expected reward. To further stabilize fine-tuning, a ratio clipping algorithm is proposed to reuse the historical data to minimize the discrepancy between new and old policies by directly constraining the difference in output action probabilities. Extensive experiments demonstrate the effectiveness and efficiency of our proposed framework in both

text-to-image and text-to-video tasks, surpassing SOTA methods in consistency and speed.

Our main innovations are as follows:

- To the best of our knowledge, we are the first to propose an initial distribution optimization framework based on policy gradients. The proposed framework is a general approach for diffusion-based generative models to produce content that is semantically closer to its input prompt.
- We formulate the optimization as a one-step MDP to efficiently adjust the initial distribution. Dynamic reward calibration module and ratio clipping algorithm are proposed to ensure the accuracy and stability of optimization.
- Extensive experiments demonstrate that our proposed work can be applied to both image and video diffusion models. Our proposed method is about an order of magnitude faster compared with SOTA. The source code will be released.

## 2 RELATED WORKS

### 2.1 Diffusion-based Generation Models

Diffusion model [14, 35] is a novel type of generative model that progressively denoise a Gaussian noise into a sample conforming to a learned data distribution by predicting the noise. Generating samples at high resolutions leads to significant computational costs for the denoising model. Latent Diffusion Model (LDM) [28] addresses this issue by utilizing a Variational Autoencoder (VAE) [16] to shift the denoising process from the pixel level to the latent space, significantly reducing computational overhead. Diffusion models have been applied across various generative tasks in different domains, achieving impressive results. These applications include image generation [7, 27, 28, 30], audio generation [9, 12, 21, 31], 3D object generation [6, 23, 29, 44], and robotics-related generation [5, 36, 42, 51] tasks. Despite their ability to generate high-quality content, these models exhibit limited control over the generated outcomes, which affects their applicability in practical scenarios.

### 2.2 Controllable Generation

To address the issue of limited control, researchers have proposed a variety of solutions. Some works leverage fine-tuning techniques to enhance models with extra conditioning layers, building upon the foundation of pretrained models. ReCo [47] and GLIGEN [20] use bounding boxes as conditional controls. SceneComposer [48] and SpaText [1] generate images by segmentation maps. ControlNet [49] is a significant contribution to this field, introducing a parallel network alongside the U-Net architecture. Beyond segmentation maps, ControlNet [49] is capable of processing various types of input, including depth maps, normal maps, canny maps, and so on. Several other works, such as Uni-ControlNet [50], UniControl [25], and T2I-Adapter [24], similarly integrate various conditional inputs to control the generation. To save on computational costs, some approaches [4, 8, 43] directly control the generation of objects during the inference phase through attention maps. These control methods modulate the generated content with additional inputs, but they offer no assistance in enhancing the control of the generated content through more precise prompts.

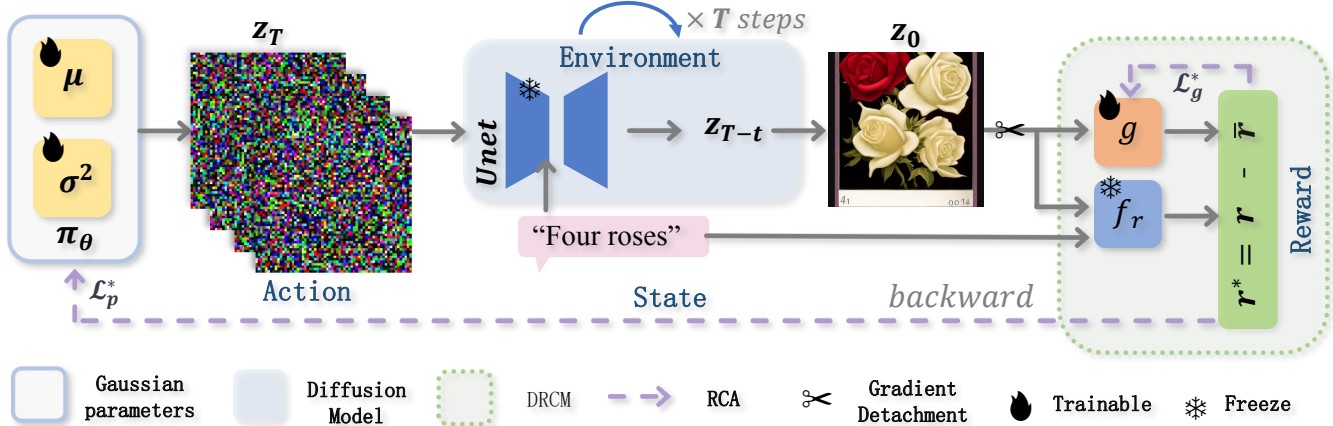

**Figure 2: The optimization iteration of our FIND. Firstly, we sample $z_T \sim \pi_\theta$, then generate an image through a T-step denoising process. Next, we optimize the reward prediction network $g$ by $\mathcal{L}_g^*$. Subsequently, we update the initial distribution $\pi_\theta$ using the policy gradient by $\mathcal{L}_p^*$.**

## 2.3 Fine-tuning Diffusion Models by Reward

Recent works [18, 41] also try to improve the alignment of text-to-image models by a reward model. The reward model is trained from a pre-trained vision-language model such as CLIP [26] or BLIP [19] by asking annotators to compare generations (learn from human feedback). Several studies [3, 10] frame fine-tuning as a multi-step decision-making process, showing that RL fine-tuning exceeds the performance of supervised fine-tuning with reward-weighted loss in reward optimization. These approaches enhance the alignment between generated content and prompts. However, the need to optimize the entire network results in high training costs and the significant fluctuations between new and old policies lead to instability during the training process. Our method only requires optimizing the initial noise distribution, significantly reducing computational overhead and our dynamic ratio clipping algorithm smoothens the policy updates, making the optimization process more stable.

## 3 PRELIMINARIES

In this section, we briefly revisit the fundamental concepts of diffusion models and the optimization objectives based on rewards.

### 3.1 Diffusion Model

Diffusion models are designed to produce high-quality, diverse content controlled by text prompts. To reduce computational costs, Rombach et al. [28] proposed a Latent Diffusion Model (LDM) that conducts the denoising process in a latent space. This model features a Variational Autoencoder (VAE) with an encoder $\mathcal{E}$ to condense the original image from pixel to latent space, and a decoder $\mathcal{D}$ to revert from latent to pixel space. The U-Net, denoted as $\epsilon_\varphi$, is involved and its structure comprises alternating down-sampling and up-sampling blocks, connected by middle blocks, each equipped with convolutional layers and spatial transformers to streamline image creation. The training of the U-Net hinges on a noise prediction loss function:

$$\mathcal{L} = \mathbb{E}_{\mathbf{z}_0, c, \epsilon \sim \mathcal{N}(0, I), t}\big[||\epsilon - \epsilon_\varphi(\mathbf{z}_t, t, \mathbf{c})||_2^2\big], \quad (1)$$

where $z_0$ is the latent code of the training sample, $\mathbf{c}$ is the text prompt condition, $\epsilon$ is the Gaussian noise, and $t$ is the time step. The noised latent code $\mathbf{z}_t$ is determined as:

$$\mathbf{z}_t = \sqrt{\bar{a}_t}\mathbf{z}_0 + \sqrt{1 - \bar{a}_t}\epsilon, \bar{a}_t = \prod_{i=1}^{t} a_t, \quad (2)$$

where $a_t$ is a hyper-parameter used for controlling the noise strength based on time $t$.

Sampling from a diffusion model initiates by selecting a random vector $\mathbf{z}_T \sim \mathcal{N}(0, I)$, which then undergoes the reverse diffusion process $p_\theta(\mathbf{z}_{t-1}|\mathbf{z}_t, \mathbf{c})$. This procedure generates a sequence $\{\mathbf{z}_T, \mathbf{z}_{T-1}, \ldots, \mathbf{z}_0\}$, culminating in the final sample $\mathbf{z}_0$. When employing DDIM [35] as the sampling method, the reverse process is described as follows:

$$p_\varphi(\mathbf{z}_{t-1}|\mathbf{z}_t, \mathbf{c}) = \mathcal{N}(\mathbf{z}_{t-1}|\epsilon_\varphi(\mathbf{z}_t, \mathbf{c}, t), \sigma_t^2 \mathbf{I}), \quad (3)$$

where $\sigma_t^2$ is fixed timestep-dependent variance.

### 3.2 Optimization of Policy Gradient

The optimization problem of policy gradients [34] is framed within the context of a Markov Decision Process (MDP), which is defined by the tuple $(S, A, \rho_0, P, R)$, with $S$ as the state space, $A$ as the action space, $\rho_0$ indicating the distribution of initial states, $P$ as the transition kernel, and $R$ representing the reward function. At each timestep $t$, an agent observes a state $\mathbf{s}_t \in S$, chooses an action $\mathbf{a}_t \in A$, earns a reward $R(\mathbf{s}_t, \mathbf{a}_t)$, and transitions to the next state $\mathbf{s}_{t+1}$ following $P(\mathbf{s}_{t+1}|\mathbf{s}_t, \mathbf{a}_t)$. Actions are determined by adhering to a policy $\pi(\mathbf{a}|\mathbf{s})$.

In the context of MDP, the agent's interactions generate trajectories, defined as sequences of states and actions $\tau = (\mathbf{s}_0, \mathbf{a}_0, \ldots, \mathbf{s}_T, \mathbf{a}_T)$. The objective of policy optimization is to maximize the agent's expected cumulative reward over these trajectories, denoted as $\mathcal{J}_\pi$, which are sampled according to its policy:

$$\mathcal{J}_\pi = \mathbb{E}_{\tau \sim p(\tau|\pi)}\big[\sum_{t=0}^{T} R(\mathbf{s}_t, \mathbf{a}_t)\big]. \quad (4)$$

# 4 METHODS

## 4.1 Overview

In this section, we provide a detailed presentation of the proposed FIND framework. Firstly, we introduce FIND formulation which is utilized along with policy gradients to optimize our initial distribution. To ensure the accuracy of our optimization direction, we introduced DRCM. Moreover, RCA is proposed to leverage historical data and enhance the stability of our training process. The pipeline is shown in Fig.2.

## 4.2 FIND formulation

Based on the findings from DDIM [35], it becomes evident that the initial noise plays a crucial role in our final generated content. Given that the diffusion model requires a multi-step denoising process, optimizing our initial distribution through direct value-based methods is not feasible. Instead, policy gradient techniques optimize the distribution of initial noise directly based on the reward, which is determined by the consistency between the generated content and the prompt. We model the process of optimizing our initial distribution using policy gradient as a one-step MDP as follows:

$$\mathbf{s} \triangleq \mathbf{c}, \quad \mathbf{a} \triangleq \mathbf{z}_T,$$
$$r \triangleq f_r(\mathbf{c}, \mathbf{z}_0), \quad \pi_\theta(\mathbf{a}) \triangleq p_\theta(\mathbf{z}_T), \tag{5}$$

in which $\mathbf{s}$ and $\mathbf{a}$ represent the state and action, respectively. The value of $\mathbf{s}$ is a constant, specifically the input prompt $\mathbf{c}$. $\mathbf{a}$ is the initial noise $\mathbf{z}_t$. $f_r$ denotes the reward function, which is used to calculate the similarity between the generated contents and the input prompt. $p_\theta$ refers to the probability of sampling $z_T$ given $\theta$. We formulate the policy $\pi_\theta$ as follows:

$$\pi_\theta = \mathcal{N}(\mu, \sigma^2), \tag{6}$$

where $\theta = \{\mu, \sigma^2\}$. $\mu$ and $\sigma^2$ are two tensors, and their sizes are the same as that of $\mathbf{z}_T$. Each element in them represents the mean and variance of an individual Gaussian distribution, corresponding to the element in $\mathbf{z}_T$, respectively. We formulate it as $\mathbf{z}_T(j) \sim \mathcal{N}(\mu(j), \sigma(j)^2)$, where $j$ is the index of the element in $\mathbf{z}_T$. We target $\pi_\theta$ as our optimization goal and employ the method of policy gradients to refine it. The term $\pi_\theta(\mathbf{a})$ refers to the probability of sampling action $\mathbf{a}$ under the policy $\pi_\theta$. DDIM [35] is utilized as our denoising strategy, ensuring that once the initial noise is specified, the resultant denoised image remains constant. Consequently, the entire T-step denoising process is treated as our environment. As shown in the blue part of Fig.2, the U-Net is frozen and does not participate in the backward process.

Specifically, the (i+1)-th optimization iteration unfolds as follows: Firstly, action $\mathbf{z}_t$ is sampled from the policy $\mathcal{N}(\mu_i, \sigma_i^2)$ which is optimized i times. Then, $\mathbf{z}_t$ is denoised to $\mathbf{z}_0$ by the environment. The reward function $f_r$ assigns a reward based on the generated content $\mathbf{z}_0$ and state $\mathbf{c}$. We use this reward to optimize the policy $\pi_\theta$ via policy gradients [34]. This optimization process is repeated multiple times until the reward is maximized. Our objective is to maximize the expected reward:

$$\arg\max_\theta \mathbb{E}_{\pi_\theta} f_r(\mathbf{c}, \mathbf{z}_0). \tag{7}$$

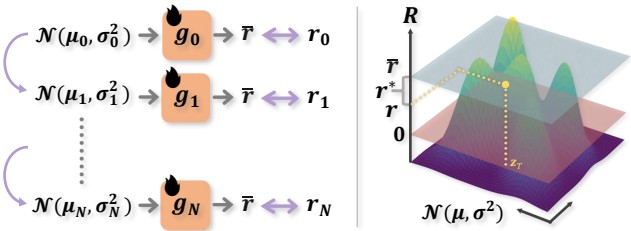

Figure 3: Left: The optimization of $g$. $N$ is the number of iterations. Right: The motivation of DRCM. $R$ is the value of reward.

To optimize this objective function, we express it in the form of gradients:

$$\nabla_\theta \mathbb{E}_{\pi_\theta}[f_r(\mathbf{c}, \mathbf{z}_0)] = \mathbb{E}_{\pi_\theta}[f_r(\mathbf{c}, \mathbf{z}_0)\nabla_\theta \log \pi_\theta]. \tag{8}$$

We present the proof of Eq.8 in Appendix A.1.

## 4.3 Dynamic Reward Calibration Module

According to the theory of policy gradient [34], the optimization direction is determined by the reward. Theoretically, if the generated content matches the prompt, the reward should be positive; otherwise, it should be negative. Since the $f_r$ is a pre-trained model, zero is not the dividing line for the quality of the reward. As illustrated in right part of Fig.3, we observe that although the reward value of our sampled yellow point is greater than 0 (indicated by the red plane), it is less than the expected reward of the current initial distribution (indicated by the blue plane). Considering this sampling point as a positive reward for optimizing the initial distribution is incorrect. The distance to the expected reward $\bar{r}$ of $\pi_\theta$ is what we required.

A straightforward approach is to estimate $\bar{r}$ through a large amount of samplings, which results in significant time consumption. We propose a Dynamic Reward Calibration Module (DRCM) to predict $\bar{r}$ of $\pi_\theta$ by a simple 3-layer MLP network $g$ defined as $\bar{r} = g(\theta)$. The loss function of $g$ is formulated as:

$$\mathcal{L}_g = ||\bar{r} - \mathbb{E}_{\pi_\theta} r||_2^2. \tag{9}$$

However, due to the lack of a ground truth dataset for the distribution and the expected reward, it is difficult to pre-train $g$. Considering the $N$ times multiple optimization steps involved in the entire process, we optimize $g$ using the rewards $r$ corresponding to the sampled $\mathbf{z}_T$ at each optimization step, as well as the initial distribution, as shown in the left part of Fig.3. We reformulate the loss function of $g$ as follows:

$$\mathcal{L}_g^* = \frac{1}{m}\sum_{k=1}^m ||\bar{r} - r^k||_2^2, \tag{10}$$

where $m$ is the number of samples in the current optimization iteration. Considering the efficiency of optimization, here $m$ is set to 1. We define the optimized reward for our current sample as $r^* = r - \bar{r}$, as the difference between the reward obtained from sampling and the reward predicted by the network. As shown in the right part of Fig.3, the yellow sampling point is treated as a negative reward for optimization by DRCM. $r^*$ is then used in Eq.7 to optimize our initial distribution.

**Algorithm 1** Initial Noise Distribution Optimize Algorithm

1: **Input**: Reward model $f_r$, text prompt $\mathbf{c}$, batch size $b$
2: Initialize $\pi_\theta = \mathcal{N}(0, \mathcal{I})$
3: **while** $\theta$ not converge **do**
4:     Obtain $b$ i.i.d. samples by first sampling $\mathbf{z}_T \sim \pi_\theta$
5:     $\mathbf{z}_0 \leftarrow \text{DDIM\_Backward}(\mathbf{z}_T, \mathbf{c})$
6:     $r \leftarrow f_r(\mathbf{z}_0, \mathbf{c})$
7:     $\bar{r} \leftarrow g(\theta)$
8:     optimize $g$ by Eq.10
9:     $r^* \leftarrow r - \bar{r}$
10:    Optimize $\theta$ by Eq.13
11: **end while**
12: **output**: Optimized initial distribution $\pi_\theta$

## 4.4 Ratio Clipping Algorithm

When using Eq.7 to optimize the initial distribution, the update process is limited to the data samples that are currently sampled. The requirement for multiple denoising steps significantly slows down the diffusion model's inference time, resulting in suboptimal optimization efficiency when repeated sampling is necessary. Further, optimizing the initial distribution solely based on the feedback from the reward function without any constraints compromises the generative performance of the original diffusion model. This issue arises because the diffusion model is trained with initial noise sampled from a standard normal distribution. If our optimized distribution deviates too far from the initial distribution, it creates a gap between the training and generation processes. We propose the Ratio Clipping Algorithm (RCA) to limit the extent of each optimization step by the historical data. Inspired by TRPO [34], we employ importance sampling, which enables the network to incorporate historical data into its updates, thereby enhancing the overall efficiency of the optimization process. We reformulate Eq.7 to a loss function as follows:

$$\mathcal{L}_p = -\mathbb{E}_{\pi_{\theta_{\text{old}}}}\left[r^* \frac{\pi_\theta}{\pi_{\theta_{\text{old}}}}\right], \tag{11}$$

where $\pi_{\theta_{\text{old}}}$ is the policy of the previous step. We formulate Eq.11 in a gradient form:

$$\nabla_\theta \mathcal{L}_p = \mathbb{E}_{\pi_{\theta_{\text{old}}}}\left[r^* \frac{\pi_\theta}{\pi_{\theta_{\text{old}}}} \nabla_\theta \log \pi_\theta\right]. \tag{12}$$

We present the proof of Eq.12 in Appendix A.2.

After establishing Eq.11, we optimize the diffusion network's initial noise distribution by leveraging historical data. Distinct from DPOK [10], which utilizes the KL divergence from the initial model to moderate the extent of parameter updates to avoid too much deviation from the original model. Our RCA, inspired by the findings of sDPO [15], adopts $\pi_{\theta_{\text{old}}}$ as the reference model. This is based on the insight that comparing parameters with those from the previous step provides a more effective upper bound for updating parameters. Specifically, we define the ratio of new policy and old policy as $\eta = \frac{\pi_\theta}{\pi_{\theta_{\text{old}}}}$. When the new policy is equal to the old policy, $\eta$ is equal to 1. To limit the magnitude of updates to the new policy, we set a margin $\lambda$, ensuring that $\eta$ falls within the range of $[1 - \lambda, 1 + \lambda]$.

We reformulate Equation 10 as follows:

$$\mathcal{L}_p^* = \begin{cases} \mathcal{L}_p, & \text{if } \eta \in [1 - \lambda, 1 + \lambda] \\ 0. & \text{else} \end{cases} \tag{13}$$

Our entire optimization process is outlined in Algo.1.

## 5 EXPERIMENTS

### 5.1 Experimental Setups

We utilize Stable Diffusion v1.5 [28] and ModelScpoe [38] as our image and video generation base model, whose parameters remain frozen throughout our optimization process. Our optimization targets are the mean $\mu$ and variance $\sigma^2$ of the initial distribution, whose sizes are 4x64x64 for image generation and 4x16x64x64 for video generation. Compared to optimizing the entire model, this approach significantly reduces the computational cost of optimization. For the reward model, we employ ImageReward[44] trained on a large dataset with human judgments for image generation and ViCLIP [39] for video generation. $\lambda$ is set to 0.02. The number of total optimization steps $N$ is 150. The learning rate is set to 0.001 and the optimizer is AdamW. All our experiments are conducted on a single 3090 GPU and the VRAM consumption is less than 10GB for image generation and 15GB for video generation.

### 5.2 Comparison with Others

To verify the effectiveness of our method, we conduct both qualitative and quantitative experiments. We compare the proposed method with the standard Stable Diffusion v1.5 [28]. Additionally, we also compare our approach with state-of-the-art approaches DPOK [10] that optimize the entire U-Net using Reinforcement Learning to highlight the efficiency and effectiveness of optimizing the initial noise.

**Quality Results.** Following the similar setting from DPOK [10], we select four prompts, *A green dog is running on the grass, A dog and a cat, Four pandas, A dog on the moon*, for fair comparison. As shown in Fig.4, in the color aspect, we can see that the baseline model generates clear images of dogs but struggles with unusual colors. DPOK generates green dogs, but in the first column, the appearance of the dog is blurred, possibly due to finetuning the network, which weakened its generative performance. In the second column, the dog generated by DPOK is not entirely green. Our method generates not only green dogs but also dogs with a very complete appearance. In terms of composition, the baseline struggles to generate multiple subjects in one image. Using DPOK, the generated images of cats and dogs exhibit some overlap, impacting the quality of the generation. Our method is capable of combinations of multiple subjects. In the counting aspect, the baseline method and DPOK struggle to generate multiple subjects of the same type in one image while our method achieves higher completeness. In terms of location, for unusual positions, the baseline method struggles and only generates common ones, such as an astronaut on the moon. The DPOK method generates dogs on the moon, but the clarity is not high. Our method is capable of generating dogs with brighter colors and higher completeness.

**Quantity Results.** In this section, we validate our proposed method from two perspectives: generative capability and computational cost. For assessing the quality of generation, we employ two metrics:

**Figure 4: Quality comparison results on different methods. The input prompt of the first two columns:** *A green dog is running on the grass.* **Third and fourth column:** *A dog and a cat.* **Fifth and sixth column:** *Four pandas.* **Seventh and eighth:** *A dog on the moon.*

ImageReward [45] and Aesthetic Score[32]. ImageReward evaluates the alignment between the generated images and the prompts. The Aesthetic Score assesses the aesthetic quality of the generated images. To validate our results, we conduct tests using prompts in four different aspects as quality evaluation. For each prompt, we select 100 images for evaluation. As shown in Tab.1, we observe that our method has advantages in terms of ImageReward, which assesses text-image alignment, particularly in the aspects of Color, Composition, and Location. In terms of Count, our method demonstrates only a slight discrepancy compared to DPOK. This highlights our method's significant advantage over both the baseline and previous SOTA methods in aligning text and images, demonstrating that we unleash the potential of the pre-trained model. In terms of Aesthetics, our method shows advantages in Composition and Count, but overall, the difference between our method, the baseline, and DPOK is minimal. This indicates that our approach, while optimizing for control effects as dictated by the prompts, does not negatively impact generative performance; instead, it may even enhance it.

**Versatility.** Followed by the experiment setting of DPOK, we evaluate our method on prompts from Drawbench [30] (10 images per each prompt). As shown in the bottom of Tab.1, our method outperforms the baseline and DPOK as well in a larger set of prompts evaluation settings. This demonstrates the comprehensive generative capability of our proposed method.

**Time Consumption.** As shown in Tab.2, our method achieves approximately 13 times faster over DPOK, completing optimization in around 14 minutes for a single prompt. This demonstrates the practicality of our approach.

## 5.3 Ablation Study

We conduct ablation studies by utilizing two complex prompts: *A red book and a yellow vase.* and *oil portrait of Batman holding a picture of Spiderman, intricate, elegant, highly detailed, lighting,*

**Table 1: Quantity results on baseline method and SOTA method and ours. Both ImageReward and Aesthetic Score are such that higher values indicate better performance.**

|  |  | ImageReward | Aesthetic |
|---|---|---|---|
| Color | Baseline | -1.64 | 5.30 |
|  | DPOK | 0.75 | **5.65** |
|  | Ours | **1.45** | 5.56 |
| Composition | baseline | 1.17 | 5.49 |
|  | DPOK | 1.16 | 5.47 |
|  | Ours | **1.43** | **5.63** |
| Count | Baseline | 0.61 | 5.70 |
|  | DPOK | **0.90** | 5.53 |
|  | Ours | 0.89 | **5.90** |
| Location | Baseline | -1.34 | **5.74** |
|  | DPOK | 0.74 | 5.21 |
|  | Ours | **1.21** | 5.61 |
| Drawbench | Baseline | 0.13 | 5.31 |
|  | DPOK | 0.38 | 5.35 |
|  | Ours | **0.39** | **5.38** |

**Table 2: The total time of optimization and inference**

|  | Baseline | DPOK | Ours |
|---|---|---|---|
| Time(min) | 0.09 | 183.3 | 13.8 |

**Table 3: The quantity results of ablation study.**

|  | Baseline | w/o DRCM | w/o RCA | Ours |
|---|---|---|---|---|
| ImageReward | -0.38 | 1.53 | 1.48 | **1.64** |
| Aesthetic | 5.72 | 5.71 | 5.98 | **6.16** |

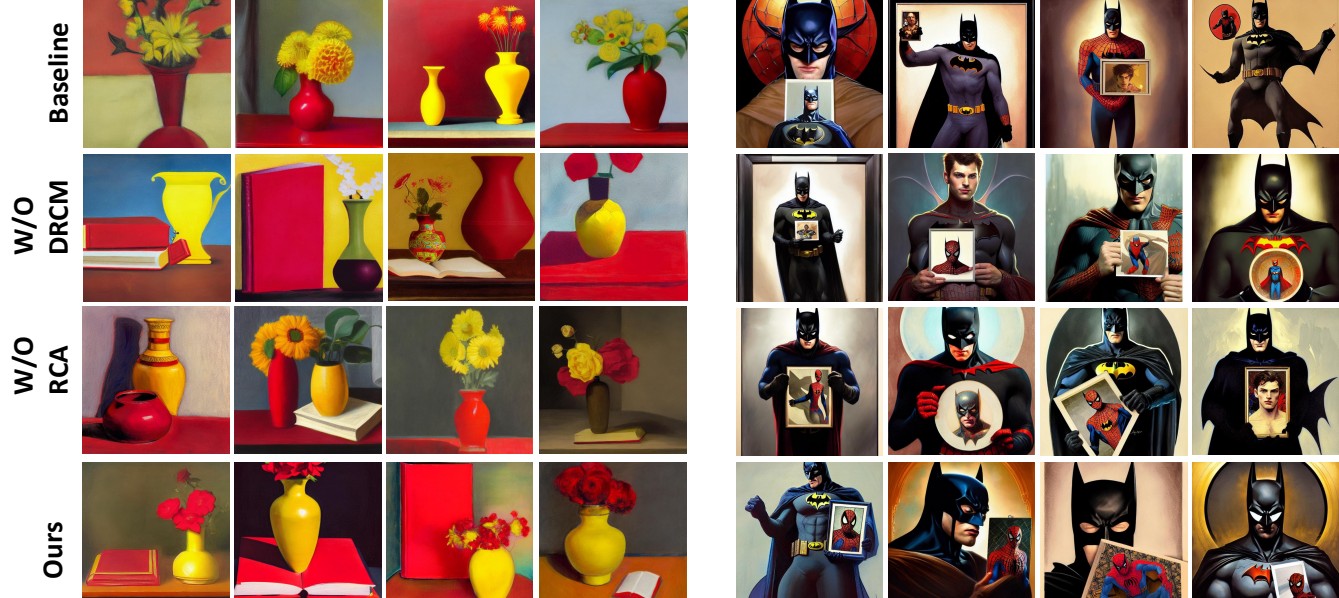

**Figure 5: Quality results of ablation study. The prompt of left part:** *A red book and a yellow vase.* **Right part:** *oil portrait of Batman holding a picture of Spiderman, intricate, elegant, highly detailed, lighting, painting, art station, smooth, illustration, art by Greg Rutkowski and Alphonse Mucha.*

*painting, art station, smooth, illustration, art by Greg Rutkowski and Alphonse Mucha.* We generate 100 samples for each scenario to serve as our test dataset.

**Impact of DRCM.** The DRCM primarily predicts the expected reward value under the current initial distribution, aiming to prevent the network from optimizing in incorrect directions. As illustrated on the left side of the second row in Fig.5, removing the DRCM leads to generated vases and books similar to the baseline, but there's a noticeable discrepancy between the colors of the vases and books and the user-input prompts. The first column shows multiple books, the vase in the second column is not yellow, the third column produces multiple vases, and in the fourth column, books turn into a table. On the right side of the second row, we observe that the generated Batman has some features of Spiderman, and the Spiderman image appears somewhat blurred. As shown in Tab.5, removing the DRCM results in a decline in both ImageReward and Aesthetic Score metrics. This is attributed to the absence of calculated expected rewards, relying solely on the sign of the reward to determine the direction of optimization leads to suboptimal solutions.

**Impact of RCA.** As demonstrated on the left side of the third row in Fig.5, the first column transforms a red book into a red bowl, the second column morphs the concept of a red book into a red vase and a yellow book, the third column generates only a red vase, and the fourth column changes the vase's color to brown. On the right half of the third row, although Batman is well generated, the spider picture he holds is poorly generated, and in the third column, although the content of the painting is well generated, the shape of the painting has turned into a trapezoid. From Tab.3, we observe a significant decrease in the ImageReward metric after

removing the RCA. This decline may be attributed to the absence of clipping operations, which means that anomalies during training cause substantial variations in the initial settings, thereby affecting the stability of the training process.

## 5.4 User Study

We recruited 33 voters from social media to assess the advantages of our method. We compared the Baseline model, DPOK, and our method. Each model generated two images from prompts in four categories same as in Sec. 5.2, Color, Composition, Count, and Location, resulting in a total of eight images. Voters were presented with 24 images and their corresponding prompts. Each image was evaluated on two dimensions: Appearance and Alignment, with scores ranging from 1 to 5, from low to high. As indicated in Tab.4, our method outperforms both the Baseline model and DPOK in all aspects. Furthermore, it's evident that users significantly prefer our method for its text-image consistency. It's also noteworthy that our method is nearly 13 times faster than DPOK.

## 5.5 Generalization for Video Diffusion

Our approach is theoretically applicable to any diffusion-based method, whether it be text-to-image, text-to-video, text-to-3D, and so forth. To demonstrate the versatility of our method, we use

**Table 4: The results of user study.**

|  | Baseline | DPOK | Ours |
|---|---|---|---|
| Quality | 2.94 | 3.27 | **3.88** |
| Alignment | 1.58 | 4.15 | **4.70** |

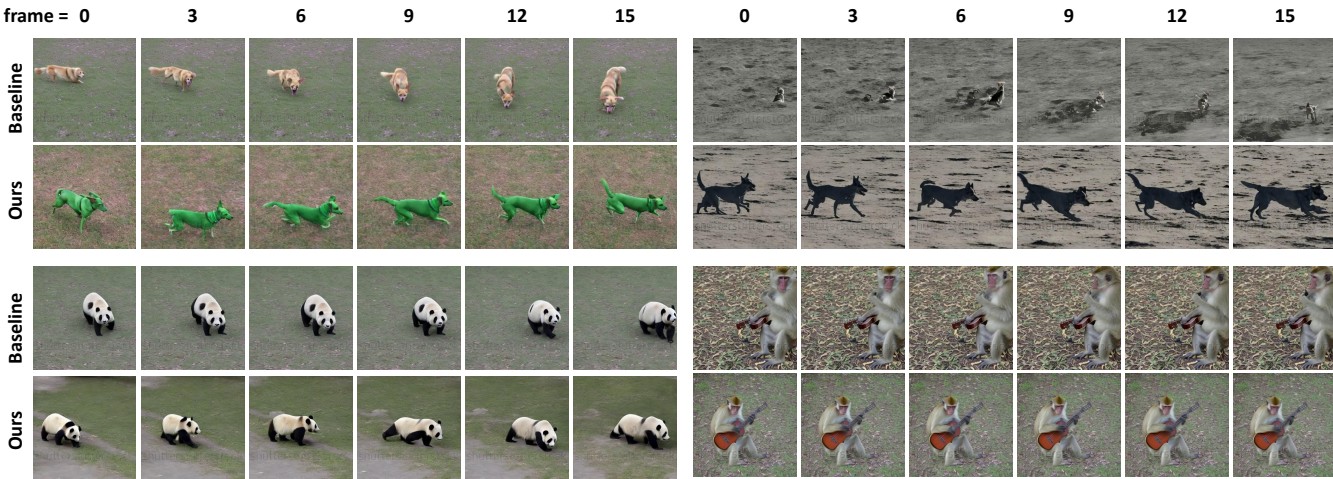

**Figure 6: Quality results on video diffusion models. The prompt of the top left corner:** *A green dog is running on the grass.* **Top right corner:** *A dog is running on the moon.* **Bottom left corner:** *A panda is walking on the grass, from left to right.* **Bottom right corner:** *A monkey is playing guitar.*

text-to-video as a case study, analyzing its performance both qualitatively and quantitatively. Specifically, we employ ModelScope [38] as our baseline model, which is a large-scale text-to-video diffusion model trained on large-scale datasets [2, 33, 46]. ViCLIP [39] is a pre-trained model used to evaluate the similarity between text and video, which is utilized as our reward function.

**Quality Results.** To verify the effectiveness of our method, we selected four sets of prompts that the baseline models struggle to generate directly: *A green dog is running on the grass.*, *A dog is running on the moon.*, *A panda is walking on the grass, from left to right.* and *A monkey is playing guitar.* These cover unusual colors, displacement control, anomalous positions, and abnormal behaviors. As illustrated in the top left of Fig.6, generating dogs with colors that do not exist in real life proves to be challenging and the dogs generated remain yellow-brown. After optimization with our method, the color of the generated dogs matches the green specified in the prompt. As illustrated in the bottom left, it is challenging for baseline models to control the objects' motion trajectories directly through prompts, leading to objects moving randomly. The panda generated by the baseline model merely turns to the right. In contrast, our method allows the panda to smoothly move from the left to the right side of the screen as requested by the prompt. In the top right corner, the quality of generated objects in anomalous positions is compromised, making the dogs appear blurry. After our optimization, the generated dogs are much clearer, and their movement process becomes smoother. As depicted in the bottom right corner, despite the ability to generate objects engaged in abnormal motions, such as monkeys and guitars, there is no interaction between them. Following our optimization, the model accurately generates behaviors such as monkeys playing guitars.

**Quantity Results.** We select four prompts same as the quality evaluation, producing 100 videos for each prompt to serve as our test dataset. ViCLIP [39] is selected to evaluate the text-video consistency of the generated videos. Following the methodology of the LOVEU-TGVE competition [40], we employ the CLIP score [13]

**Table 5: The quantity results on video diffusion.**

|  | ViCLIP | Consistency | PickScore |
|---|---|---|---|
| Baseline | 0.21 | 0.83 | 20.08 |
| Ours | **0.28** | **0.88** | **21.29** |

to assess the consistency between frames. Additionally, PickScore [17] is used to predict user preferences for our model. As shown in Tab.5, our method surpasses the baseline across all three metrics, demonstrating that the videos generated after optimization with our approach have improved in terms of text-video consistency, inter-frame consistency, and predicted user preferences.

## 6 CONCLUSION

In this paper, we introduced FIND, a novel **F**ine-tuning **I**nitial **N**oise **D**istribution with policy optimization framework, to align the content generated by diffusion models with user-input prompts. Unlike previous methods that required extensive training or additional controls, our approach was capable of optimizing for any prompt within just 13 minutes. We observed that the initial noise significantly influences the final output of diffusion models, leading us to optimize the initial noise. However, optimizing the initial distribution from the generated images was challenging due to the multi-step denoising required by diffusion models. We utilized policy gradients to circumvent the multi-step denoising, optimizing the initial distribution directly through the reward function. To ensure that the optimization direction was not solely determined by the sign of the reward, we proposed the DRCM to predict the expected value of the reward under the current distribution. Additionally, we developed the RCA module to leverage past samples and ensured optimization stability. Both quantitative experiments and qualitative tests have proven the effectiveness of our proposed method. Moreover, our approach can be applied to kinds of diffusion-based generative models, demonstrating its high generalizability and versatility.

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

Received 20 February 2007; revised 12 March 2009; accepted 5 June 2009

