# OpenReview forum: "FIND: Fine-tuning Initial Noise Distribution with Policy Optimization for Diffusion Models"
_acmmm.org/ACMMM/2024/Conference — MM2024 Poster_

### Official Review · Reviewer_tYkj · 2024-05-24

**Rating:** 3
**Confidence:** 3

**Summary:**

This paper proposes an algorithm, FIND, to enhance the alignment between input prompts and generated samples in diffusion-based text-to-image/video generation models. FIND refines the initial noise distribution of a given input prompt using a policy gradient algorithm. Additionally, the authors apply the dynamic reward calibration module and ratio clipping to improve the optimization performance.

**Strengths:**

1. FIND algorithm optimizes the initial noise distribution for each input prompt instead of the diffusion model parameter. This changes the multi-step MDP setting into a single-step MDP setting, which might reduce the optimizing difficulties.

2. The qualitative samples and quantitative scores look better the baselines.

**Limitations:**

I have several concerns:
1. The FIND algorithm relies on per-prompt optimization of the initialization distribution, which raises concerns about sampling efficiency. According to Table 3, it takes approximately 14 minutes to fine-tune a single prompt, which seems excessive given that generating a single image with a diffusion model typically only takes a few seconds. In contrast, DPOK[1] learns the information into the model parameters and can be fine-tuned on multiple prompts simultaneously (section 5.5 of [1]). The finetuned model from [2] can be also applied to different prompts. Therefore, it might not be fair to directly compare the optimization times of FIND and DPOK, as DPOK can work on various prompts after fine-tuning, whereas FIND is only suitable for one specific prompt. The lengthy optimization time for a single prompt could limit FIND's applicability in many real-world applications.

2. FIND approximates the initial distribution for a given prompt as a simple Gaussian distribution. I am uncertain whether this approximation is valid. If we consider this distribution as a policy, then this straightforward policy form needs to capture the landscape formed by the full diffusion process and the reward model, which seems to be very complex. Specifically, I have the following questions:

(a) After the distribution is fitted, can different samples from the same initial distribution generate different valid images/videos corresponding to the same prompt, or do they collapse into a single mode of nearly identical samples? Figure 4 shows different samples for the same prompt, but I'm not sure whether these samples come from different instances of the same fitted initial distribution or from different fitted initial distributions for the same prompt.

(b) Similar to question (a), under different random seeds, will the algorithm converge to the same initial distribution, or will it capture different modes?

(c) Failure case study: Given different random seeds and prompts, can FIND always find a valid initialization? Given an initialization distribution, are all samples from this distribution valid, or will there be cases where some samples are valid while others are not?

3. Besides building the sampling policy upon the initialization distribution, FIND essentially follows the same reinforcement learning framework as [2]. The ratio clipping algorithm is also very similar to the formulations of TRPO [3] and PPO [4].

[1] DPOK: Reinforcement Learning for Fine-tuning Text-to-Image Diffusion Models.
[2] Training diffusion models with reinforcement learning.
[3] Trust region policy optimization.
[4] Proximal policy optimization algorithm.

**Suitability:**

3

---

### Official Review · Reviewer_JAqv · 2024-05-24

**Rating:** 4
**Confidence:** 3

**Summary:**

This paper proposes a method to fine-tune the initial noise of the conditional diffusion model to improve the consistency between prompts and the generated image. This method outperforms SOTA methods and reduces the generation time.

**Strengths:**

This paper reveals that the initial noise of diffusion models can be controlled to tune the generated image. The initial noise is optimized using a reinforcement learning method, which is novel and interesting.

**Limitations:**

1) It seems that the network $g$ needs to be trained for each prompt. I am a bit concerned about the time required for generating large-size images. In Table 2, the size of the images is not mentioned.

2) It would be better to have a comparison of the memory usage. For generating high-resolution images, it seems like the network $g$ requires a huge memory. Assume the size of images as 256$\times$256, The input size of network $g$ will be $393216=2\times3\times256\times256$. The network $g$ is a 3-layer MLP. If the first layer outputs features with a size of 393, the weight of the first layer will be $w\in\mathbb{R}^{393216\times 393}$ which is huge. Can authors provide more detail about network $g$?

**Suitability:**

3

---

### Official Review · Reviewer_LtJH · 2024-05-24

**Rating:** 4
**Confidence:** 2

**Summary:**

This paper addresses the challenges of aligning generated content with user prompts in diffusion-based generative models. The authors introduce the FIND framework, which optimizes the initial noise distribution using policy optimization to enhance consistency between generated content and user prompts. The paper thoroughly examines the limitations of existing methods, highlighting the difficulties in achieving prompt-consistent generation and the challenges in directly optimizing the initial noise distribution.

**Strengths:**

1.The paper addresses a highly relevant and timely topic by introducing the FIND framework, which focuses on optimizing the initial noise distribution of diffusion models to enhance the consistency between generated content and user prompts. This topic is highly relevant in current generative model research.
2.The paper provides a comprehensive examination of this topic, including the analysis of reasons for inconsistencies between generated content and prompts in existing models, the design of the framework, and experimental validation. The content is detailed and rich.
3.The introduction of policy optimization into the initial noise distribution optimization framework and the proposal of dynamic reward calibration module and ratio clipping algorithm to ensure stability and accuracy during the optimization process is particularly noteworthy for its innovative approach.
4.The manuscript is well-referenced, indicating that the authors have conducted an in-depth review and analysis of existing research in the fields of diffusion models, controllable generative models, and policy optimization.
5.The authors' application of the one-step Markov decision process to optimize the initial noise distribution and the extensive experiments demonstrating the effectiveness and superiority of the method in both image and video generation tasks are commendable. The proposed method significantly outperforms existing state-of-the-art methods in achieving consistency between generated content and prompts, with a training speed improvement of an order of magnitude.

**Limitations:**

1.Although the paper proposes the idea of improving generative performance by optimizing the initial noise distribution, the explanation of this motivation and its significance is not sufficiently in-depth. Readers might need more background information to understand why existing methods do not adequately address this issue and what the unique advantages of this approach are.
2. Although the paper conducts extensive experiments, they are primarily focused on text-to-image and text-to-video generation tasks. It could be further extended to other generative tasks, such as 3D object generation, to demonstrate the broader applicability of the method.
3. The proposed method in the paper includes multiple modules (such as the dynamic reward calibration module and the ratio clipping algorithm). While these modules help with optimization, they also add complexity to the method. The paper does not adequately discuss the impact of these modules on the overall system complexity and computational cost.
4. The paper does not provide sufficiently detailed descriptions of the optimization process and parameter settings, especially the implementation details of the policy gradient optimization. This may lead to difficulties for readers attempting to reproduce the work.

**Suitability:**

2

---

### Meta-Review · Area_Chair_xKEH · 2024-07-02

**Recommendation:** Accept (Poster)
**Confidence:** 5

**Metareview:**

The reviewers agreed that this paper proposes novel and interesting research. LtJH calls the work “highly relevant and timely”. This reviewers extensive list of strengths includes that the paper is detailed in terms of overall examination of the topic, that it is well-referenced, that the proposed method is “noteworthy for it’s innovative approach” and “outperforms state of the art methods”.

2/3 reviewers suggest that this work is “definitely suitable” and I agree that it is very appropriate work for ACM MM.

In terms of limitations, LtJH suggests further exploration of the motivation and extension of experiments to other media, e.g., 3D models.

Tykj notes several questions related to the approximation of intial distribution, and the potential for generating a good range of outputs from one prompt.

All reviewers noted issues with the time required to train the noise distribution for each prompt and this was still a significant issue following the rebuttal.

tYkJ puts it well: “According to Table 3, it takes approximately 14 minutes to fine-tune a single prompt, which seems excessive given that generating a single image with a diffusion model typically only takes a few seconds. In contrast, DPOK[1] learns the information into the model parameters and can be fine-tuned on multiple prompts simultaneously (section 5.5 of [1]).”

This issue is certainly discussed in the rebuttal and I felt that the response was reasonable given that this is an unconventional approach.

My feeling is that, although this length of time is unusually long compared to present diffusion models, this unique idea has shown effectiveness and may inspire other modifications to the diffusion process. Although 14 minutes per prompt is impractical today, this good idea may last for a long time in the literature. I feel that we should support interesting, appropriate albeit unusual ideas to support intellectual diversity at MM.
TYKJ gave me permission to not worry about this concern and so given the mix of ratings, I will lean towards acceptance, particularly given that this is a very suitable topic for MM.